# Tumor Intrinsic Immunogenicity Suppressor SETDB1 Worsens the Prognosis of Patients with Hepatocellular Carcinoma

**DOI:** 10.3390/cells13242102

**Published:** 2024-12-19

**Authors:** Chang-Qing Yin, Chun-Qing Song

**Affiliations:** 1College of Life Sciences, Zhejiang University, Hangzhou 310058, China; yinchangqing@westlake.edu.cn; 2Key Laboratory of Growth Regulation and Translational Research of Zhejiang Province, School of Life Sciences, Westlake University, Hangzhou 310024, China; 3Westlake Laboratory of Life Sciences and Biomedicine, Hangzhou 310024, China; 4Laboratory of Gene Therapeutic Biology, Institute of Basic Medical Sciences, Westlake Institute for Advanced Study, Hangzhou 310024, China

**Keywords:** SETDB1, hepatocellular carcinoma, immune cell infiltration, prognosis

## Abstract

Hepatocellular carcinoma (HCC) is clinically distinguished by its covert onset, rapid progression, high recurrence rate, and poor prognosis. Studies have revealed that SETDB1 (SET Domain Bifurcated 1) is a histone H3 methyltransferase located on chromosome 1 and plays a crucial role in carcinogenesis. Therefore, we aimed to evaluate the clinical significance of SETDB1 expression in HCC. In patients with HCC, elevated levels of SETDB1 correlated with a poorer overall survival (OS) rate, marking it as an independent prognostic factor for HCC, as revealed by both univariate and multivariate Cox analyses. Furthermore, we utilized the SangerBox and TISIDB databases to profile the tumor immune microenvironment in HCC, including scoring the tumor microenvironment and assessing immune cell infiltration. The TIDE algorithm was employed to examine the association between SETDB1 expression and immune responses. Our findings indicated that SETDB1 expression negatively correlated with the majority of immune cells, a wide range of immune cell marker genes, and numerous immune pathways, thereby leading to the reduced effectiveness of immune checkpoint inhibitors. Lastly, both in vivo and ex vivo experiments were conducted to substantiate the role of SETDB1 in HCC tumorigenesis. In conclusion, the upregulation of SETDB1 is associated with a poorer prognosis in HCC patients and inversely correlates with immune cell infiltration, potentially serving as a predictive marker for immunotherapy response.

## 1. Introduction

Hepatocellular carcinoma (HCC), the most prevalent histological subtype of primary liver cancer, accounts for from 75% to 85% of all primary liver cancer cases and is the third leading cause of cancer-related mortality [1,2,3]. Globally, HCC ranks sixth in tumor incidence but has the third highest mortality rate, with a 5-year survival rate of less than 10% [4,5]. Despite advancements in traditional therapies such as surgery, interventional therapy, chemotherapy, and immunotherapy, postoperative relapse, early metastasis, and progressive drug resistance remain significant factors contributing to the poor prognosis of HCC patients [6]. Therefore, there is an urgent need to develop more reliable diagnostic biomarkers and innovative indicators to predict HCC patients’ survival and facilitate personalized treatment strategies.

SETDB1, a member of the SET family of histone-lysine N-methyltransferases, plays a crucial role in regulating embryonic stem cell development and maintaining the developmental state of these cells [7,8,9,10]. Additionally, SETDB1 has been implicated in promoting osteoblast proliferation and guiding cell fate decisions [11,12]. Previous studies have shown that abnormally high levels of SETDB1 stimulate the proliferation, migration, and invasion of tumor cells in various cancers, including breast cancer, colorectal cancer, gastric cancer, ovarian cancer, prostate cancer, and lung cancer [13,14,15,16,17,18,19]. In humans, SETDB1 is located at 1q21.3 and was identified as the most significantly upregulated epigenetic modulator in HBV-related HCC through RNA-seq [20]. The mechanisms leading to its upregulation in HCC are multifaceted, including its increased susceptibility to copy number gains due to its location on chromosome 1, its regulation by the SP1 transcription factor, and its targeting by certain miRNAs [20,21]. The primary objective of our study was to investigate the relationship between SETDB1 expression and the clinicopathological parameters of HCC, as well as its potential as a prognostic factor.

The tumor microenvironment (TME), which is primarily composed of tumor-infiltrating immune cells (TIICs), such as neutrophils, CD4^+^ helper T cells, CD8^+^ cytotoxic T cells, NK cells, and macrophages, has been identified as a critical factor influencing the progression and treatment of HCC [22,23]. A wealth of evidence suggests that TIICs impact the biological behavior of HCC cells and ultimately affect patients’ prognosis [24]. Additionally, numerous studies and clinical trials have demonstrated that the overall survival (OS) and disease-free survival (DFS) of HCC patients treated with Immune Checkpoint Blockade (ICB) have significantly improved, providing patients with the possibility of subsequent surgical treatment [25,26,27]. SETDB1 has been previously identified as an epigenetic checkpoint that suppresses tumor-intrinsic immunogenicity by repressing latent transposable element (TE)-derived regulatory elements [28,29]. However, the potential of SETDB1 to promote HCC progression, influence HCC TIICs, or act as a predictor of ICB treatment response has not yet been clarified.

Based on transcriptomic data from The Cancer Genome Atlas (TCGA) and the Gene Expression Omnibus (GEO), as well as in vivo and ex vivo experiments, this study investigated the expression of SETDB1 and its influence on the malignant biological behavior of HCC, as well as its correlation with the clinical characteristics, prognosis, and immune cell infiltration. Our findings suggest that SETDB1 plays a carcinogenic role in HCC and may serve as a potential prognostic predictor.

## 2. Materials and Methods

### 2.1. Data Acquisition

The flowchart of this study, depicted in Appendix A, delineates the data analysis process. Gene sequence data and clinical information for 374 HCC samples were obtained from the TCGA database (https://cancergenome.nih.gov/, accessed on 7 November 2023). We also downloaded HCC transcriptional profiling data from GSE14520 [30,31] with two different platforms in the GEO (https://www.ncbi.nlm.nih.gov/, accessed on 7 November 2023). The original data were extracted into matrix files using Perl software (Version 5.38.0), and information about human gene names was downloaded from the Ensembl website (https://asia.ensembl.org/index.html, accessed on 7 November 2023).

### 2.2. Analysis of SETDB1 Expression in HCC

SETDB1 expression in pan-cancer was identified by TIMER 2.0 (http://timer.cistrome.org/, accessed on 9 November 2023) [32]. We also evaluated SETDB1 expression in HCC tumor tissues and normal liver tissues using immunohistochemistry data from the HPA database (http://www.proteinatlas.org/, accessed on 9 November 2023) and UALCAN (http://ualcan.path.uab.edu/, accessed on 9 November 2023). Additionally, the intracellular localization of SETDB1 was investigated in mouse primary hepatocytes and Hepa1-6 cells. Patients were divided into two groups based on the median value of SETDB1 expression as the cut-off point, and we compared the differences in clinical characteristics between the two groups.

### 2.3. The Correlation Between SETDB1 Expression and HCC Prognosis

Kaplan-Meier survival analysis was performed to investigate the relationship between SETDB1 expression and the prognosis of HCC patients using GEPIA 2.0. Additionally, univariate and multivariate Cox regression analyses were conducted to determine whether SETDB1 could be considered as an independent prognostic factor.

### 2.4. GeneMANIA, STRING Analyses and Functional Annotation

The GeneMANIA (https://genemania.org/, accessed on 11 November 2023) database was used to demonstrate the connections of other genes to SETDB1. We also investigated the protein–protein interaction (PPI) network of SETDB1-related genes using the STRING (www.string-db.org/, accessed on 11 November 2023) database. The top 300 SETDB1-related genes were identified using GEPIA 2.0. The biological functions of SETDB1 in HCC were predicted using Metascape (https://metascape.org/, accessed on 11 November 2023). Subsequently, differential expression analysis was performed with a threshold set at |log2FC| > 1 and adjusted *p* < 0.05. The significantly differentially expressed genes (DEGs) were then annotated using GO and KEGG analyses.

### 2.5. The Correlation Between SETDB1 and Immune Microenvironment

We investigated the correlation between SETDB1 and tumor microenvironment scores, including Immune score, Stromal score, and ESTIMATE score, using the Sangerbox database (http://vip.sangerbox.com/home.html/, accessed on 15 November 2023) [33]. Given the importance of immune cells in the tumor microenvironment, we subsequently explored the relationship between SETDB1 and the abundance of immune cell infiltration using the TISIDB database (http://cis.hku.hk/TISIDB/, accessed on 15 November 2023). Additionally, since the Tumor Immune Dysfunction and Exclusion (TIDE) integrates the characteristics of T cell dysfunction and exclusion, it has prominent advantages over other biomarkers [34]. Therefore, we utilized the TIDE score from the TIDE database (http://tide.dfci.harvard.edu/, accessed on 15 November 2023) to predict the potential immunotherapy effect.

### 2.6. Mouse Primary Hepatocytes Isolation

Primary mouse hepatocytes were isolated using a multi-step collagenase procedure via the vena cava, followed by 50% percoll gradient purification, as previously described [35]. The isolated primary hepatocytes were seeded in collagen-coated 6-well plates and cultured in William’s medium E with GlutaMAX (35050061, ThermoFisher, Plainville, MA, USA) and 1% Penicillin-Streptomycin.

### 2.7. Cell Culture and Infection

The HEK293T cells and mouse HCC cell lines Hepa1-6 from ATCC were cultured in DMEM medium supplemented with 10% FBS in a humidified incubator with 5% CO_2_ at 37 °C. HEK293T cells were used to package lentivirus-encoding individual sgRNA and Cas9. Single-guide RNA (sgRNA) sequences were designed as described [35]. The sgSETDB1 (5′-CCAAGGGAAGCGAAGATCAT-3′) and sgControl (5′-CGAGGTATTCGGCTCCGC-3′) oligos (Tsingke Biotech, Hangzhou, China) were annealed and cloned into the pX330 vector (addgene 42230) or lentiV2 (addgene 52961) using standard BbsI or BsmBI protocols. CRISPR/Cas9 lentivirus packaging particles were generated in the HEK293T cells. Hepa1-6 cells were seeded in 6-well plates and then infected with SETDB1 CRISPR/Cas9 lentivirus and control lentivirus. Twenty-four hours later, puromycin was added to the medium. Subclones of virus-transduced Hepa1-6 cells were selected and expanded.

### 2.8. Immunofluorescence Staining

For immunofluorescence staining, fixed primary mouse hepatocytes, and Hepa1-6 cells were permeabilized with 1% Triton-X-100 and then blocked with 10% goat serum. The glass coverslips were incubated with anti-SETDB1 (MA5-15722, ThermoFisher, Plainville, MA, USA) overnight at 4 °C. After incubation with a fluorescent antibody (A-11001, ThermoFisher, Plainville, MA, USA), the glass coverslips were counterstained with DAPI and observed under a fluorescence microscope.

### 2.9. The Construction of the HCC Mouse Model

Four- to six-week-old FVB mice were purchased from Westlake University (Hangzhou, China). The plasmids, including px330-U6-sgP53 (mouse) 20 μg, pT3-EF1α-c-Myc (human) 5 μg, and pCMV-sleeping beauty transposase 2 μg, were harvested using the EndoFree Plasmid Maxi Kit (12362, Qiagen, Düsseldorf, Germany) and delivered into the FVB mice via hydrodynamic tail vein injection (HDT) as previously described [35]. The mice were sacrificed 4 weeks later when they developed burdens of HCC. All procedures involving animals were conducted in strict accordance with the guidelines approved by the Institutional Animal Care and Use Committee of Westlake University.

### 2.10. Immunohistochemistry

Tissues were fixed in 4% formalin overnight and then embedded in paraffin. Then, 5 μm liver sections were subjected to HE staining or probed with antibodies using standard immunohistochemistry (IHC) protocols. The antibodies used included anti-SETDB1 (11231-1-AP, Proteintech, Rosemont, IL, USA) and anti-MYC (Ab32072, Abcam, Fremont, CA, USA).

### 2.11. Colony Formation Assay

A total of 500 Hepa1-6 cells stably expressing non-target control or sgSETDB1 were plated in 6-well plates and grown for two weeks. The cells were then fixed with 4% formaldehyde and stained with 1% crystal violet (A100528, BBI life sciences, Shanghai, China). The colonies were counted and analyzed.

### 2.12. In Vivo Subcutaneous Xenograft Tumor Model

In brief, 1 × 10^6^ Hepa1-6 cells stably expressing non-target control or sgSETDB1 were resuspended in 100 µL DMEM and injected subcutaneously into the right flank of 4- to 6-week-old male FAH/Prkdc/IL2RG mutation mice. The mice were sacrificed 14 days later, and the tumors were collected.

### 2.13. Western Blotting

Total protein lysates were extracted from Hepa1-6 cells using RIPA buffer (P0013C, Beyotime, Shanghai, China) and quantified using the BCA method (P0012, Beyotime, Shanghai, China). A total of 20 μg of proteins were probed with primary antibodies, including SETDB1 (11231-1-AP, Proteintech, Rosemont, IL, USA), MYC (Ab32072, Abcam, Fremont, CA, USA) and GAPDH (60004-1-Ig, Proteintech, Rosemont, IL, USA).

### 2.14. Statistical Analysis

All statistical analyses were performed in R 3.6.2 software, IBM SPSS Statistics 23.0, and online databases. Kaplan–Meier curves were created to analyze the relationship between survival time and SETDB1 expression level. The correlations between SETDB1 and immune checkpoints and tumor microenvironment scores were analyzed by Spearman’s correlation analysis. A *p* < 0.05 was considered statistically significant.

## 3. Results

### 3.1. SETDB1 Expression Is Upregulated in HCC

To investigate the possible role of SETDB1 in tumor pathogenesis, we analyzed the gene expression of SETDB1 in several human cancers from TCGA. As shown in Figure 1A, SETDB1 expression was significantly upregulated in several tumor tissues, including bladder urothelial carcinoma (BLCA), breast invasive carcinoma (BRCA), cholangiocarcinoma (CHOL), colon adenocarcinoma (COAD), esophageal carcinoma (ESCA), glioblastoma multiforme (GBM), head and neck squamous cell carcinoma (HNSC), kidney chromophobe (KICH), kidney renal clear cell carcinoma (KIRC), hepatocellular carcinoma (HCC), lung adenocarcinoma (LUAD), and lung squamous cell carcinoma (LUSC). Data from TCGA-HCC indicated higher SETDB1 expression in HCC compared to normal tissues (*p* < 0.001) (Figure 1B,C). Additionally, datasets from the GEO database, GSE14520-GPL3921 (Figure 1D) and GSE14520-GPL571 (Figure 1E), also showed that SETDB1 expression was higher in HCC compared to normal liver tissues. Data from the CPTAC database further confirmed that the protein levels of SETDB1 were increased in HCC compared to normal tissues (Figure 1F).

Next, to further confirm the expression of SETDB1 in HCC, we compared the SETDB1 expression level using the IHC staining from the HPA database. The results showed that SETDB1 expression was higher in HCC tissues compared to normal tissues (Figure 2A). The subcellular localization of SETDB1 was examined in mouse primary hepatocytes and Hepa1-6 cells using immunofluorescence. The results indicated that SETDB1 is primarily localized in the nucleus (Figure 2B).

### 3.2. Correlation Between SETDB1 Expression and the Prognosis of HCC Patients

The correlation between clinicopathological characteristics and SETDB1 expression is presented in Table 1. Notably, the expression level of SETDB1 was significantly correlated only with gender. HCC patients with high SETDB1 expression had poorer OS compared to those with low SETDB1 expression (*p* = 0.029) (Figure 2C). However, there was no significant difference in disease-free survival (DFS) between the low and high SETDB1 expression groups in HCC patients (*p* = 0.16) (Figure 2D). Univariate Cox regression analysis showed that the factors that affect the survival of HCC patients include T-stage (*p* < 0.001, HR = 2.126), M-stage (*p* = 0.017, HR = 4.077), pathologic stage (*p* < 0.001, HR = 2.090), and SETDB1 expression level (*p* = 0.008, HR = 1.342) (Table 2). Furthermore, multivariate Cox regression analysis revealed that the factors independently affecting the survival of HCC patients were M-stage (*p* = 0.033, HR = 3.642) and SETDB1 expression level (*p* = 0.016, HR = 1.366) (Table 2). These findings suggest that SETDB1 may be an independent factor for predicting the prognosis of HCC patients.

### 3.3. Experimental Verification of the SETDB1 Function in HCC

As the overexpression of the oncogene *Myc* is commonly observed in HCC [35,36,37], and approximately 30% of human HCC samples exhibit *Myc* gene amplification [38]; we validated the SETDB1 expression levels by knocking out *p53* and overexpressing *Myc* by HDT in FVB mice (Figure 3A). The HCC model formed multiple tumors (Figure 3B). As expected, the liver tumors had obviously increased SETDB1 expression compared to normal liver tissues (Figure 3C). IHC staining results show that SETDB1 colocalized well with the MYC protein (Figure 3D). To validate the functional role of SETDB1 in HCC cells, we established a stable SETDB1 knockout cell line in Hepa1-6 cells using a sgRNA sequence. Successful knockout of SETDB1 was confirmed (Figure 3E). We found that stable knockout of SETDB1 significantly suppressed the proliferation of Hepa1-6 cells, as demonstrated by the cell clone formation assay (Figure 3F). To further verify the effect of SETDB1 knockout on HCC tumorigenicity in vivo, we constructed a subcutaneous xenograft tumor model. The results showed that the stable knockout of SETDB1 in Hepa1-6 cells significantly reduced the size of HCC tumors formed in the FAH/Prkdc/IL2RG mutation mice compared with the non-target sgRNA control (Figure 3G).

### 3.4. SETDB1 Interaction Network and Functional Analysis in HCC

Firstly, the GeneMANIA database was used to establish a network of SETDB1-associated genes, including ATF7IP, CHD7, CLK2, ARL14EP, MPHOSPH8, SUCLG1, ZNF274 and TASOR (Figure 4A). Furthermore, STRING database analysis showed that SETDB1 interacted with HNRNPA0, HNRNPU, RBMX, PTBP1, and SFPQ (Figure 4B). The results from the MetaScape tool showed that the potential function of SETDB1 in HCC was associated with mRNA splicing, chromatin organization, histone modification, regulation of the cell cycle process, and recombinational repair (Figure 4C). DEGs were often used to explore the potential biological role by enrichment analysis. To further confirm the potential biological role of SETDB1 in HCC, we performed functional enrichment analysis on TCGA transcriptome data. The DEGs between the SETDB1 high group and SETDB1 low group were then compared, with the criteria set of absolute |log2FC| > 1 and an adjusted *p* < 0.05. The heatmap showed the hierarchical clustering analysis of these DEGs (Figure 5A), and the corresponding volcano plot is shown in Figure 5B. The specific gene names can be found in Appendix A. To gain deeper insights into the function of SETDB1, KEGG and GO enrichment analyses were conducted. According to the KEGG analysis, upregulated DEGs were mainly involved in pathways such as Herpes Simplex virus 1 infection, neuroactive ligand–receptor interaction, nicotine addiction, glutamatergic synapse, calcium signaling pathway, and African trypanosomiasis (Figure 5C). GO analysis of the upregulated DEGs revealed that most of these genes were associated with processes such as pattern specification, homophilic cell adhesion via plasma membrane adhesion molecules, cell–cell adhesion, regionalization, glutamate receptor signaling pathway, limb development, appendage development, tryptophan catabolic process, and indole-containing compound catabolic process (Figure 5D). Meanwhile, the results of KEGG and GO analyses of downregulated DEGs are shown in Figure 5E,F. The above results showed that SETDB1 may be involved in HCC pathogenesis through multiple signaling pathways and metabolic processes.

### 3.5. The Correlation of SETDB1 with HCC Immune Characteristics

Numerous studies have underscored the critical influence of the TME in HCC tumorigenesis and progression. We evaluated the ESTIMATE Score, Immune Score, and Stromal Score of HCC using an estimation algorithm, and analyzed the correlation between these three scores with the expression level of SETDB1. The results showed that SETDB1 expression was negatively correlated with Immune score (*p* = 3.9 × 10^−4^, Cor = −0.19) (Figure 6A), Stromal score (*p* = 1.8 × 10^−6^, Cor = −0.25) (Figure 6B), and ESTIMATE score (*p* = 9.2 × 10^−6^, Cor = −0.23) (Figure 6C). We utilized the TISIDB database to analyze the relations between SETDB1 expression and the abundance of 28 Tumor-infiltrating lymphocytes (TILs). As illustrated in Figure 6D, the relationships between SETDB1 expression and TILs in different types of cancer were exhibited. Specifically, in HCC, SETDB1 expression was negatively correlated with CD8^+^ cells (r = −0.229, *p* = 8.48 × 10^−6^), Th1 cells (r = −0.469, *p* < 2.2 × 10^−16^), NK cells (r = −0.349, *p* = 5.65 × 10^−12^), and monocytes (r = −0.453, *p* < 2.2 × 10^−16^) (Figure 6E).

### 3.6. The Benefit of ICB Therapy in HCC Subgroups

Immune Checkpoint Blockade (ICB) therapy, combined with first-line chemotherapy, can be used as a novel tumor therapy, outperforming traditional standard chemotherapy and improving the prognosis of patients with multiple types of cancer [39,40,41]. Next, we investigated the correlation between SETDB1 expression and several common immune checkpoint genes. Interestingly, in HCC, SETDB1 expression was negatively correlated with eight common immune checkpoint markers, including CD96, CD244, CD274, CSF1R, HAVCR2, LGALS9, PDCD1LG2, and TIGIT (Figure 7A). Notably, CD274, also known as PD-L1, is a target of immune checkpoint pathways. These findings led us to hypothesize that the SETDB1 gene may play a crucial role in tumor immunity. We then evaluated the potential therapeutic efficacy of immunotherapy in various HCC subgroups using TIDE. The higher the TIDE prediction score is, the more likely immune escape will occur, indicating that patients with a high TIDE score have a lower response to ICB therapy. As shown in Figure 7B–D, in HCC, the SETDB1-high subgroup had a higher T cell dysfunction score, T cell exclusion score, and TIDE score than the SETDB1-low subgroup (*p* < 0.001). Our results indicated that HCC patients with high SETDB1 expression may have a poorer response to ICB therapy and a worse prognosis. These results demonstrate that SETDB1 has good predictive value for the response of HCC patients to ICB therapy.

## 4. Discussion

Although some cancer treatment modalities have been implemented, there is still limited effectiveness and a poor prognosis for HCC patients. As a promising strategy to treat various cancers, cancer immunotherapy has shifted the paradigm of cancer treatment in recent years, often outperforming traditional methods in terms of effectiveness [42]. Current approaches in cancer immunotherapy for HCC focus on immune checkpoints, cancer vaccines, and combination therapies, including chemotherapy and radiotherapy [43]. Notably, immune checkpoint blockade (ICB) therapy, involving anti-PD-1, anti-CTLA-4, and anti-PD-L1 antibodies, fundamentally differs from traditional antitumor chemotherapy. Since ICB does not target normal tissue cells, its side effects are significantly reduced. By altering the interaction between immune cells and tumor cells, as well as the tumor microenvironment, ICB can stimulate immune cells to attack tumors, achieving the ultimate goal of tumor treatment [44,45,46,47]. However, the low response rate and the tendency to develop drug resistance are significant bottlenecks currently faced by this therapy [39]. Therefore, there is an urgent need to identify novel immune infiltration-related biomarkers to characterize patient prognosis and explore the underlying mechanisms of HCC pathogenesis.

Previous research has established SETDB1 as an epigenetic checkpoint that suppresses the intrinsic immunogenicity of tumors [28]. When SETDB1 is lost, transposable elements (TEs) that have the potential to encode viral proteins are de-repressed, leading to the generation of MHC-I peptides and the activation of T-cell responses [28]. However, whether SETDB1 can serve as a candidate target for immunotherapy in HCC has not been fully elucidated. Therefore, our purpose was to identify the role of SETDB1 as a predictor of outcomes in HCC patients and to explore the biological functions and potential regulatory pathways of SETDB1 in HCC using a comprehensive bioinformatics analysis. The mechanism for SETDB1 becoming the most significantly upregulated epigenetic regulator in HCC may involve multiple steps, including recurrent SETDB1 gene copy gains at chromosome 1q21 and enhanced activity by the hyperactivation of the SP1 transcription factor [20]. Additionally, at the post-transcriptional level, it is also a direct target of certain miRNAs [20,21]. To investigate the underlying biological function of the SETDB1 gene, we identified a series of DEGs by comparing the two groups based on SETDB1 expression and performed GO and KEGG functional enrichment analyses. The results suggest that SETDB1 may promote tumor invasion and migration and help the tumor cells escape from immune surveillance through cell–cell adhesion and interaction-related biological processes. Additionally, SETDB1 regulates the evolution of immune genes within segmental duplications and may maintain epigenetic memory by silencing TEs with gene-regulatory functions [48]. Blocking SETDB1 in a mouse cancer model significantly increased the cell-killing activity of immune cells against cancer cells, and the tumors shrank significantly after treatment with an immune checkpoint inhibitor (PD-1 blockade), suggesting the potential of SETDB1 as a promising target for immunotherapy.

Cancer cells have rich interactions with various stromal cells in the tumor microenvironment (TME), including actively recruiting specific stromal cells into the tumor tissue and altering the stromal cell state and extracellular matrix (ECM) composition with the tumor [49,50]. Additionally, tumor-infiltrating immune cells (TIICs) are essential components of the TME and are capable of monitoring cancer cells and influencing the prognosis of cancer patients [32,51]. According to the TISIDB database, we identified a negative association between SETDB1 expression and TIICs in the TME. Moreover, SETDB1 expression was significantly associated with the Immune Score, Stromal Score, and ESTIMATE Score in HCC. Interestingly, a negative correlation was also found between expression of SETDB1 and CD274 (PD-L1). Therefore, blocking SETDB1 may upregulate PD-L1 expression, potentially mediating immune evasion through the PD1/PD-L1 pathway. However, the combination of SETDB1 and PD-L1 inhibition may achieve a superior anti-tumor effect: targeting SETDB1 to directly reduce tumor cell survival and targeting PD-L1 to counteract the negative effects of SETDB1 inhibition and enhance immune function. Overall, our research addresses an unmet clinical need by combining anti-PD-L1 therapy with SETDB1 inhibition, which may offer a potential approach for HCC therapy.

Although our study shows initial promise, it is important to note that it is currently an in silico analysis and has limitations in the functional validation of our results. These limitations include the use of only a *Myc*-induced HCC mouse model, the lack of measurement of transposable element activity in HCC samples, and the absence of clear evidence regarding the regulation of PD-L1 by SETDB1 in HCC. Despite these limitations, our findings provide a foundational step toward exploring the potential of SETDB1 in translational applications. In an era where molecular profiling of HCC from TCGA data is nearly complete, translating these findings into clinical practice remains a significant challenge. Future studies are necessary to establish the molecular relationships between SETDB1 suppression and the identified pathways, further validating the therapeutic potential of targeting SETDB1 in HCC.

## 5. Conclusions

In conclusion, our results reveal that SETDB1 could predict the prognosis of cancer patients and correlate with immune infiltration levels in HCC. Therefore, the identification of SETDB1 as a new prognostic biomarker may facilitate the development of new immunotherapeutic strategies.

## Figures and Tables

**Figure 1 cells-13-02102-f001:**
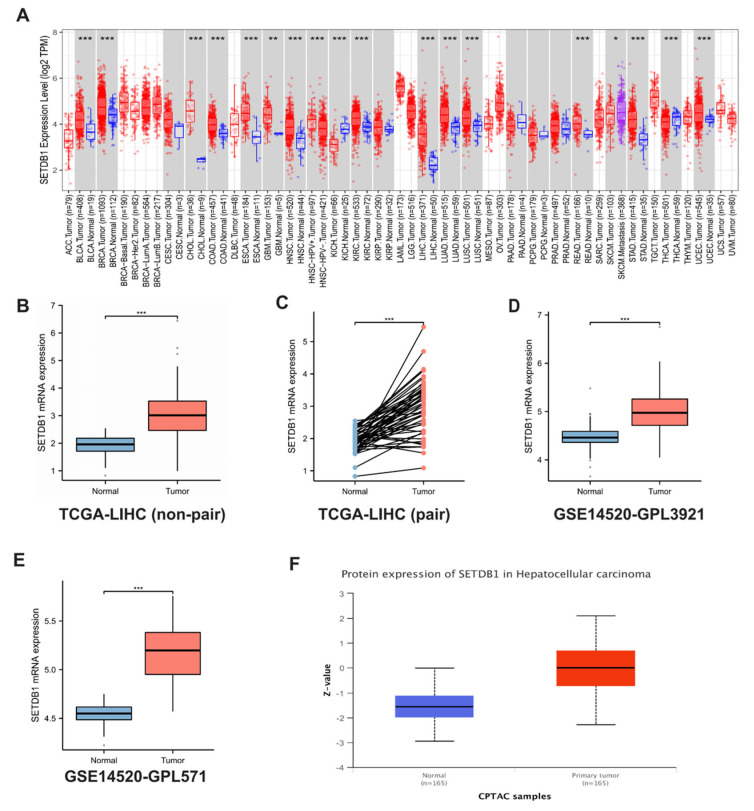
SETDB1 was upregulated in pan-cancers. (**A**) The expression of the SETDB1 gene in pan-cancers and their corresponding normal tissues from TCGA data. (**B**) The non-paired expression of SETDB1 between normal and tumor tissues. (**C**) The paired expression of SETDB1 between normal and tumor tissues. The mRNA expression of SETDB1 was higher in HCC than in normal liver tissue in the GSE-14529-GPL3921 (**D**) and GSE14520-GPL571 (**E**) datasets. (**F**) The protein levels of SETDB1 were higher in primary tumor tissues than in normal tissues in CPTAC samples. * *p* < 0.05; ** *p* < 0.01; *** *p* < 0.001.

**Figure 2 cells-13-02102-f002:**
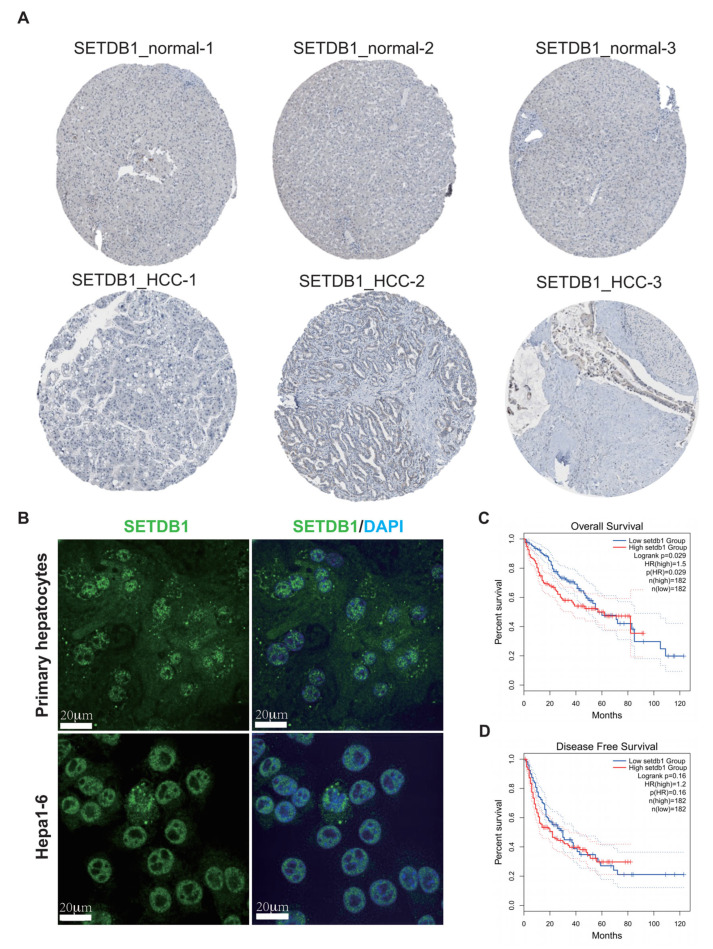
Correlation between SETDB1 expression and the prognosis of HCC patients. (**A**) The protein expression of SETDB1 was obtained from the Human Protein Atlas. (**B**) The subcellular localization of SETDB1 in mouse primary hepatocytes and Hepa1-6 cells. The correlation of SETDB1 expression with OS (**C**) and DFS (**D**) in HCC patients. Scale bar = 20 μm.

**Figure 3 cells-13-02102-f003:**
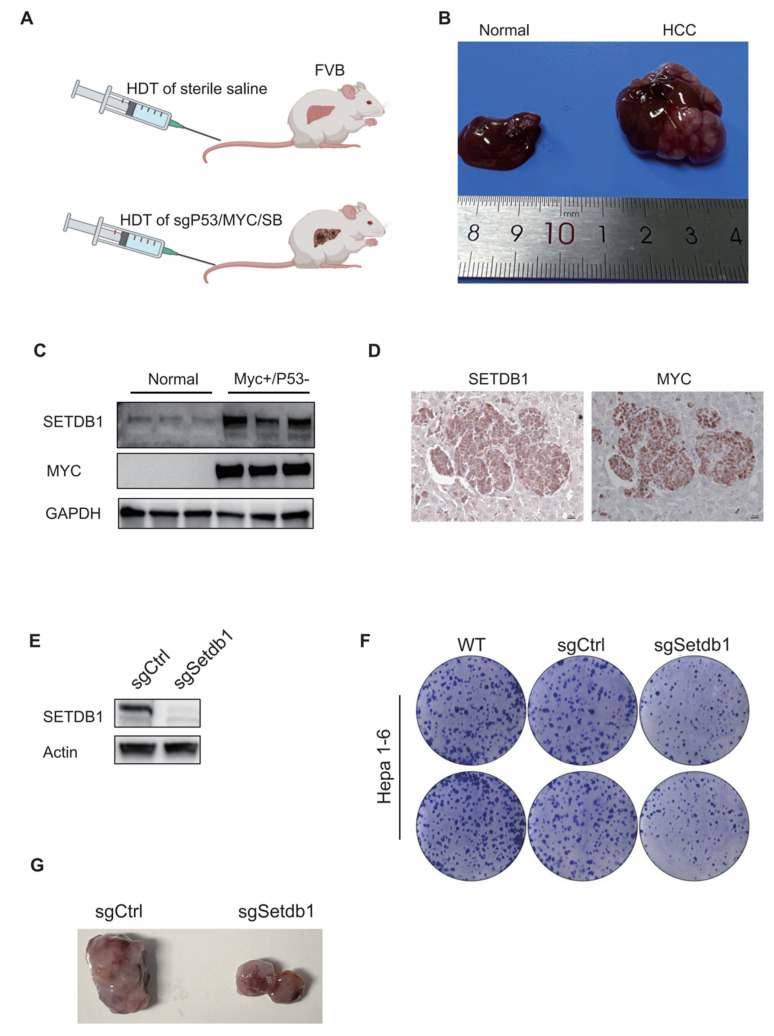
In vivo and ex vivo validation of the SETDB1 function in HCC. (**A**) HCC model design. FVB mice were injected with normal saline or sgP53/c-Myc/SB plasmids, respectively. (**B**) Representative liver tissues of normal (**left**) and HCC mice (**right**). (**C**) The protein expression of SETDB1 and MYC in the normal and HCC tissues. (**D**) The IHC staining of SETDB1 and MYC in HCC liver tissues. (**E**) The SETDB1 was successfully knocked out in Hepa1-6 cells. (**F**) The SETDB1 knockout inhibited colony formation in Hepa1-6 cells. (**G**) The SETDB1 knockout suppressed tumor formation in the subcutaneous xenograft tumor model.

**Figure 4 cells-13-02102-f004:**
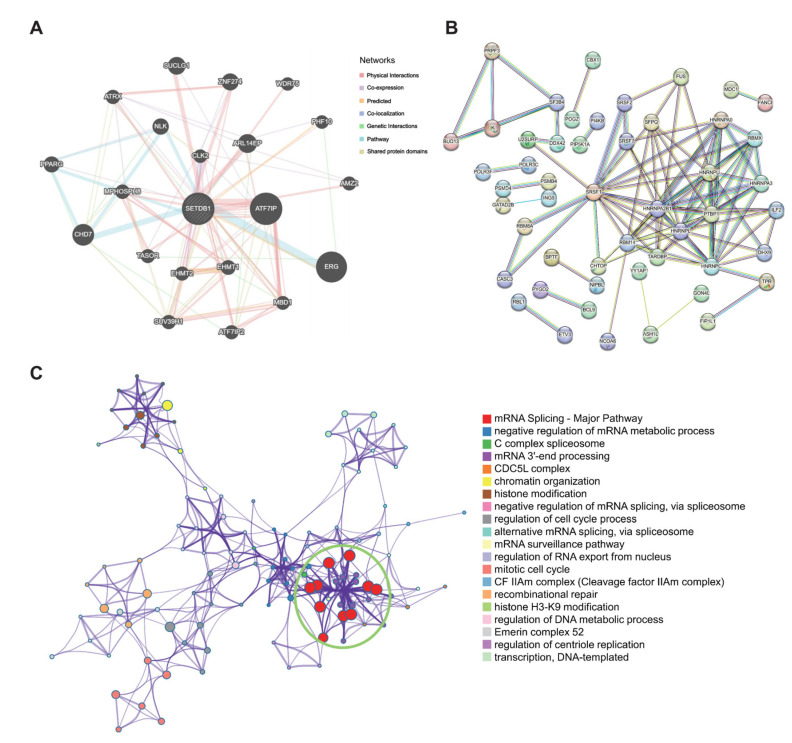
SETDB1 Interaction Network. PPI network of SETDB1 in the GeneMANIA (**A**) and STRING (**B**) tools. (**C**) Metascape analysis of SETDB1.

**Figure 5 cells-13-02102-f005:**
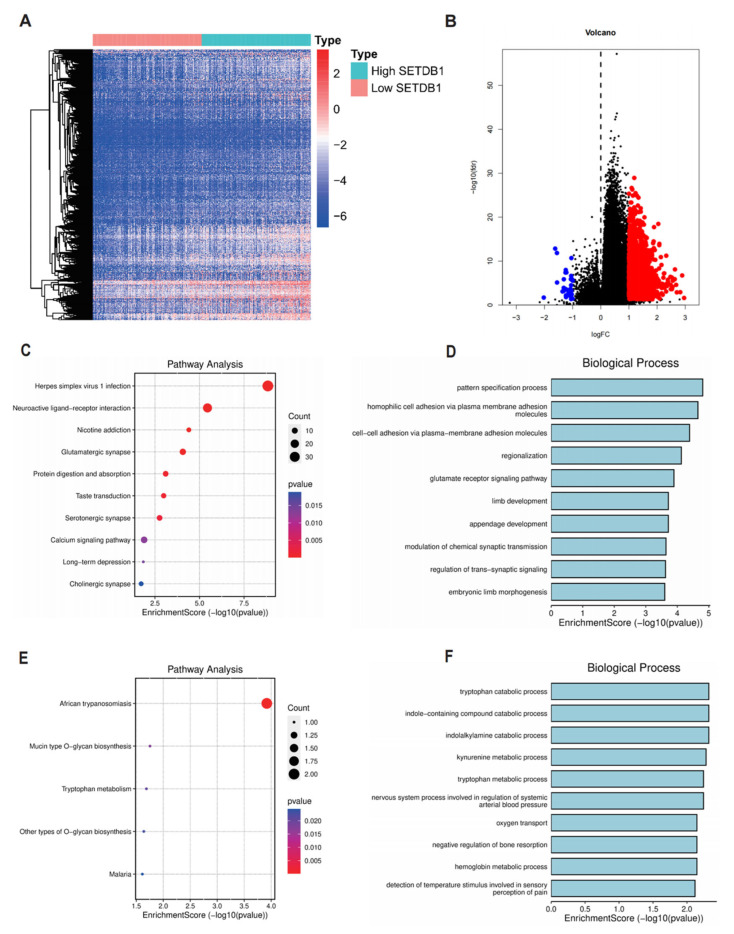
Enrichment analysis of SETDB1 expression-correlated DEGs in HCC. (**A**) Clustering analysis heatmap of SETDB1 expression-correlated DEGs. (**B**) Volcano plot of DEGs between samples with high SETDB1 expression and low SETDB1 expression. KEGG analysis (**C**) and GO analysis (biological process) (**D**) in SETDB1 expression-correlated upregulated DEGs. KEGG analysis (**E**) and GO analysis (biological process) (**F**) in SETDB1 expression-correlated downregulated DEGs.

**Figure 6 cells-13-02102-f006:**
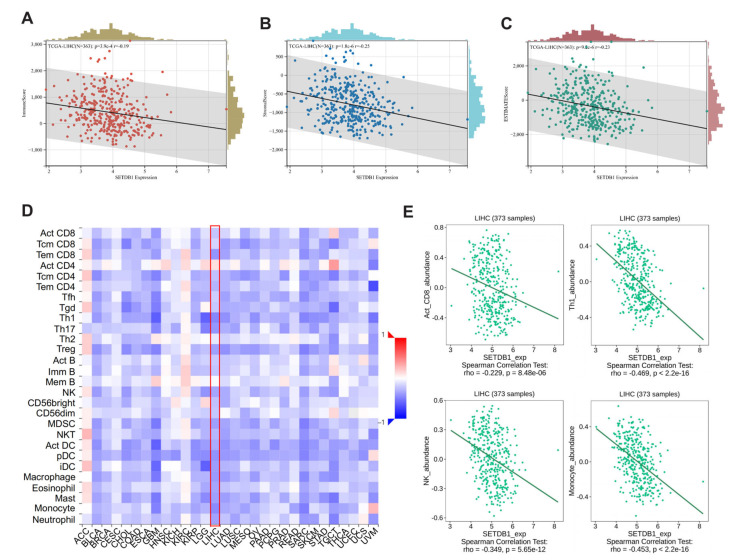
Correlation analysis between SETDB1 and immune microenvironment. Correlation of SETDB1 with Immune score (**A**), Stromal score (**B**), and ESTIMATE score (**C**) in HCC. (**D**) The landscape of the relationship between SETDB1 expression and TILs in multiple types of cancers (red denotes positive correlation, and blue denotes negative correlation). (**E**) SETDB1 expression was significantly negatively associated with infiltrating levels of act_CD8, Th1, NK, and Monocyte in HCC.

**Figure 7 cells-13-02102-f007:**
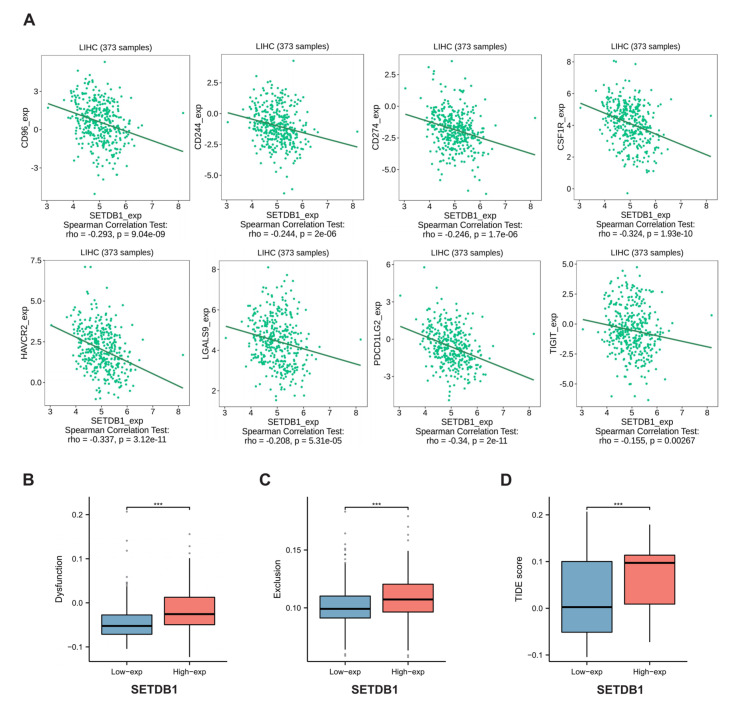
(**A**) SETDB1 expression was negatively related with CD96, CD244, CD274, CSF1R, HAVCR2, LGALS9, PDCD1LG2, and TIGIT in HCC. (**B**) Differences in the T-cell dysfunction score between the low and high expression of SETDB1 in HCC. (**C**) Differences in the T-cell rejection score between the low and high expression of SETDB1 in HCC. (**D**) Differences in the tumor immune dysfunction and exclusion (TIDE) score between the low and high expression of SETDB1 in HCC. *** *p* < 0.001.

**Table 1 cells-13-02102-t001:** Clinicopathological characteristics of the HCC patients with high and low SETDB1 expression.

Characteristic	SETDB1 Expression		*p*	Method
	Low (*n* = 187)	High (*n* = 187)		
**Age, n (%)**			0.133	Chisq.test
≤60	81 (21.7%)	96 (25.7%)		
>60	106 (28.4%)	90 (24.1%)		
**Age, median (IQR)**	63 (53, 69)	60 (51, 68)	0.111	Wilcoxon
**Gender, n (%)**			0.047	Chisq.test
Female	51 (13.6%)	70 (18.7%)		
Male	136 (36.4%)	117 (31.3%)		
**Race, n (%)**			0.177	Chisq.test
Asian	69 (19.1%)	91 (25.1%)		
Black or African American	9 (2.5%)	8 (2.2%)		
White	98 (27.1%)	87 (24%)		
**T stage, n (%)**			0.566	Chisq.test
T1	97 (26.1%)	86 (23.2%)		
T2	44 (11.9%)	51 (13.7%)		
T3	36 (9.7%)	44 (11.9%)		
T4	7 (1.9%)	6 (1.6%)		
**N stage, n (%)**			0.624	Fisher.test
N0	122 (47.3%)	132 (51.2%)		
N1	1 (0.4%)	3 (1.2%)		
**M stage, n (%)**			0.369	Fisher.test
M0	132 (48.5%)	136 (50%)		
M1	3 (1.1%)	1 (0.4%)		
**Pathologic stage, n (%)**			0.175	Fisher.test
Stage I	94 (26.9%)	79 (22.6%)		
Stage II	42 (12%)	45 (12.9%)		
Stage III	36 (10.3%)	49 (14%)		
Stage IV	4 (1.1%)	1 (0.3%)		

**Table 2 cells-13-02102-t002:** Univariate and multivariate Cox regression analyses.

Characteristics	Univariate Analysis		Multivariate Analysis	
	HR (95% CI)	*p* Value	HR (95% CI)	*p* Value
**Age**				
≤60	Reference			
>60	1.205 (0.850–1.708)	0.295		
**Gender**				
Male	Reference			
Female	1.261 (0.885–1.796)	0.2		
**Race**				
Asian	Reference			
Black or African American&White	1.341 (0.926–1.942)	0.121		
**T stage**				
T1	Reference			
T2&T3&T4	2.126 (1.481–3.052)	<0.001	0.543 (0.075–3.947)	0.546
**N stage**				
N0	Reference			
N1	2.029 (0.497–8.281)	0.324		
**M stage**				
M0	Reference			
M1	4.077 (1.281–12.973)	0.017	3.642 (1.108–11.967)	0.033
**Pathologic stage**				
Stage I	Reference			
Stage II&Stage III&Stage IV	2.090 (1.429–3.055)	<0.001	4.232 (0.571–31.349)	0.158
**SETDB1 exp**	1.342 (1.081–1.666)	0.008	1.366 (1.059–1.761)	0.016

## Data Availability

The data underlying this study are freely available from the GEO database (https://www.ncbi.nlm.nih.gov/geo/, accessed on 7 November 2023), TCGA database (https://cancergenome.nih.gov/, accessed on 7 November 2023), HPA database (http://www.proteinatlas.org/, accessed on 9 November 2023), TIMER 2.0 (http://timer.cistrome.org/, accessed on 9 November 2023), GeneMANIA (https://genemania.org/, accessed on 11 November 2023), STRING (www.string-db.org/, accessed on 11 November 2023), TISIDB database (http://cis.hku.hk/TISIDB/, accessed on 15 November 2023), Sangerbox (http://vip.sangerbox.com/home.html/, accessed on 15 November 2023), TIDE database (http://tide.dfci.harvard.edu/, accessed on 15 November 2023), and so on.

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
