# Peer review of "Tumor Intrinsic Immunogenicity Suppressor SETDB1 Worsens the Prognosis of Patients with Hepatocellular Carcinoma"

_cells, 2024, doi:10.3390/cells13242102_

Round 1
Reviewer 1 Report
Comments and Suggestions for Authors
In this manuscript, the authors reanalyze the role of SETDB1 in hepatocellular carcinoma (HCC) using the material present in the data database. They consider that SETDB1 might represent an interesting marker in HCC as it is overexpressed in most tumors.
It appears that SETDB1 is highly expressed in HCC, but this overexpression is not associated with remarkable clinical features in high or low expressers as shown in Table 1.
The original experimental contribution of the paper consists essentially of the analysis of mouse models. The authors looked for SETDB1 expression in mouse cells by Immunofluorescence. The authors also construct a mouse model of HCC based on overexpression of Myc and inactivation of tp53. The choice of myc-dependent mouse model of HCC presented as “logical” is not obvious as MYC is overexpressed in only 17% of human HCC (https://cancer.sanger.ac.uk/cosmic/gene/analysis?ln=MYC). This vision of the MYC gene overwhelmingly involved in liver carcinogenesis is outdated.
SETDB1 in man is mapping in 1q21.3, a locus that undergoes the highest rate of copy number gains in HCC. The authors did not mention this fact which might explain the frequent overexpression of the gene.
The authors do not provide the mean or median fold expression change in HCC by comparison with non-tumor livers.
Major problems:
Abstract: no figures, no P values. It is vague and therefore makes the appraisal of the phenomena described difficult.
Results:
Figure 5: Figure 5A displayed a lot of blue ie underexpressed genes, a situation compatible with SETDB1 activity but Figure 5B displays on the contrary many upregulated genes (red). Could the authors explain this apparent paradox?
Readers would be happy to see the names of differentially expressed genes on the volcano plot.
Figures 5C-F: are we looking at the GO terms of overexpressed or underexpressed genes? The functions enriched in these gene lists are probably quite different.
Discussion: It is not a discussion but rather a recap of the results. Notably, the literature about SETDB1 in HCC (18 PubMed references) is largely forgotten.
Figure 7: It is barely convincing as the correlation coefficient is <0.3 in 5 cases out of 8. 0.3 is usually considered as the minimal threshold to consider a correlation. Finding P values with such a high number of points is not unexpected even if the correlation is weak.
The authors, once again, consider that they are bringing an original idea considering SETDB1 as a valuable target in cancer. However, this idea was proposed as early as 2019 (Invention S. Amigorena). Please, consider more adequately the former literature.
Why did the authors not try to measure the activity of transposable elements (MERVs: mouse endogenous retrovirus) in their model of mouse tumors and cell lines? These elements have been described as crucial in the control of tumor immunogenicity by SETDB1 (Griffin, Nature, 2021)
Minor issues:
In the abstract please mention the residues targeted by SETDB1
Introduction: Line 49, the authors affirm the insufficiency of the literature about SETDB1 and HCC. There are 18 references in PubMed but none are mentioned.
The Gene Ontology terms characterizing differentially expressed genes in high SETDB1 expressers are puzzling and barely linked to a clear tumor process: Herpes Simplex virus 1 infection, neuroactive ligand-receptor interaction, nicotine addiction, glutamatergic synapse, calcium signaling pathway, and African trypanosomiasis.
This fact is not commented on by the authors. Could they provide us with some insights?
Furthermore, as the authors were obviously as puzzled as their readers, they used different software to understand the interactions of SETDB1 with its usual partners. This approach yielded completely different outcomes. Could the authors try to synthesize a bit the outcome of this analysis because as it is it is not intelligible?
Legend of Figure 4: STRING not SRTING
Page 13, line 310: “These findings led us to hypothesize that the SETDB1 gene may be crucial for tumor immunity. » The authors attributed to themselves the idea that SETDB1 is a controller of tumor cell immunity. But this idea was already formulated (ref 26, 2021, cited in introduction). Please, tone down.
Author Response
"Dear Reviewer, Please see the attachment."

Reviewer 2 Report
Comments and Suggestions for Authors
The manuscript of Yin and Song deals with the relationship between SETDB1, a histone lysine methyltransferase and the onset of hepatocarcinoma (HCC) playing a pivotal role in carcinogenesis. Through multivariate analysis authors found that SETDB1 correlated with a poorer overall survival (OS) rate, marking it as an independent prognostic factor for HCC; moreover, SETDB1 expression negatively correlated with the majority of immune cells and numerous immune pathways, thereby leading to an ineffectiveness of immune checkpoint inhibitors. Authors concluded that up-regulation of SETDB1 is associated with a poorer prognosis in HCC patients and inversely correlates with immune cell infiltration, thereby potentially serving as a predictive marker for immunotherapy response. This work appeared to be well performed, showing interesting data. However, I have mainly one concern:
- Authors said that we conducted a comprehensive bioinformatics analysis on SETDB1 in cancers, in order to find biological functions and potential regulatory pathways in HCC. Despite they revealed interesting interactions with other factors, I think that the SETDB1 role in liver carcinogenesis remains obscure. I suggest to authors to perform a “graphical abstract” or figure showing the mechanisms really involved in the favoring action of this protein on cancer growth.
Author Response

(The authors gave the same response as above.)

Reviewer 3 Report
Comments and Suggestions for Authors
In this manuscript, the authors evaluated the clinical significance of SETDB1 expression in HCC. They found that elevated SETDB1 levels were associated with a poorer overall survival (OS) rate. They also demonstrated that SETDB1 expression was negatively correlated with the immune response. Through both in vivo and ex vivo experiments, the authors explored the role of SETDB1 in HCC tumorigenesis and showed that the upregulation of SETDB1 is associated with a poorer prognosis in HCC patients and inversely correlates with immune cell infiltration
Several areas need improvement for better clarity.
Minor comments:
- In Figure 1A, please explain why SETDB1 expression in several normal tissues is missing.
- In Figre 2B, it seems that the SETDB1 expression is higher in primary hepatocytes that in Hep1-6. Please comment on this.
- Please describe more clearly how the immunoscore and stromal score were determined.
- There are minor grammatical errors, line 309 for example. A thorough language review is required.
Author Response

(The authors gave the same response as above.)
